# A new method based on physical patterns to impute aerobiological datasets

**Sofia Tagliaferro**[1], **Adrián Corrochano**[2], **Pierpaolo Marchetti**[1], **Alessandro Marcon**[1☯]*, **Soledad Le Clainche**[2☯]

**1** Unit of Epidemiology and Medical Statistics, Department of Diagnostics and Public Health, University of Verona, Verona, Italy, **2** School of Aerospace Engineering, Universidad Politécnica de Madrid, Madrid, Spain

☯ These authors contributed equally to this work.
* alessandro.marcon@univr.it

**Data Availability Statement:** All the materials underlying the results presented in the study are available from the Modelflows-app website (https://modelflows.github.io/modelflowsapp/airpollution/). These include the original pollen datasets

## Abstract

Limited research has assessed the accuracy of imputation methods in aerobiological datasets. We conducted a simulation study to evaluate, for the first time, the effectiveness of Gappy Singular Value Decomposition (GSVD), a data-driven approach, comparing it with the moving mean interpolation, a statistical approach. Utilizing complete pollen data from two monitoring stations in northeastern Italy for 2022, we randomly generated missing data considering the combination of various proportions (5%, 10%, 25%) and gap lengths (3, 5, 7, 10 days). We imputed 4800 time series using the GSVD algorithm, specifically implemented for this study, and the moving mean algorithm of the "AeRobiology" R package. We assessed imputation accuracy by calculating the Root Mean Square Error and employed multiple linear regression models to identify factors independently affecting the error (e.g. pollen variability, simulation settings). The results showed that the GSVD was as good as the well-established moving mean method and demonstrated its strong generalization capabilities across different data types. However, the imputation error was primarily influenced by pollen characteristics and location, regardless of the imputation method used. High variability in pollen concentrations and the distribution of missing data negatively affected imputation accuracy. In conclusion, we introduced and tested a novel imputation method, demonstrating comparable performance to the statistical approach in aerobiological data reconstruction. These findings contribute to advancing aerobiological data analysis, highlighting the need for improving imputation methods.

## Introduction

Aerobiology is a recent discipline focusing on atmospheric bioaerosols, such as pollen and spores [1, 2]. Its interdisciplinary approach allows for the examination of the impacts of climate change and the development of innovative methodologies aimed at managing allergic diseases [3]. Aerobiological data are typically measured on daily basis and are provided by local/national monitoring networks. Despite the existence of automatic sampling devices, current monitoring practices primarily rely on manual samplers, introducing the possibility of systematic errors [4–6].

downloaded from POLLnet (https://pollnet.isprambiente.it/), the R and Python codes to generate and input missing data, sample gappy datasets, a brief overview of the paper, and a video explanation of the methodology.

**Funding:** A.M. received grants to conduct the MEETOUT study from the European Union through the Italian Ministry of University and Research under the ESF REACT-EU Green and Innovation funding programme (Ministerial Decree 1061/2021) and the NextGenerationEu funding programme (Ministerial Decree 737/2021). Article processing charges were supported by the special fund at the University of Verona dedicated to Open Access publications. S.L.C. and A.C. acknowledge the grants PID2023-147790OB-I00, TED2021-129774B-C21 and PLEC2022-009235 funded by MCIN/AEI/10.13039/501100011033 and by the European Union "NextGenerationEU"/PRTR. The authors acknowledge the MODELAIR and ENCODING projects that have received funding from the European Union's Horizon Europe research and innovation programme under the Marie Sklodowska-Curie grant agreement No. 101072559 and 101072779, respectively. The results of this publication reflect only the authors view and do not necessarily reflect those of the European Union. The European Union cannot be held responsible for them. A.C. acknowledges the support of Universidad Politécnica de Madrid, under the program 'Programa Propio'. The funders had no role in study design, data collection and analysis, decision to publish, or preparation of the manuscript. There was no additional external funding received for this study.

**Competing interests:** The authors have declared that no competing interests exist.

Pollen time series are frequently incomplete due to malfunctions and maintenance of the monitoring stations [5], as well as voluntary interruptions in periods considered irrelevant for the measures. Consequently, the presence of missing data in aerobiological datasets is common, prompting the need for imputation methods. Traditional methods, as omitting to assign values to missing data, may lead to underestimation errors [1, 7, 8].

Statistical and artificial intelligence methodologies have been implemented for data imputation. Statistical approaches such as linear interpolation [9–12], cubic spline interpolation [12], the Gaussian method [13] or averaging values from other years for each day with missing data [9], are commonly used in aerobiological studies. The availability of pre-set statistical software packages facilitates the application of the most common methodologies used in data imputation [1]. Recently, the "AeRobiology" R package was developed specifically to manage and visualize aerobiological data, as well as to impute missing data [6]. In this package different interpolation methods are implemented, including linear, moving mean, spline, time series analysis, and nearby locations interpolation. The moving mean method is a statistical univariate approach. It consists of filling in missing values by averaging nearby data within a symmetrical interval that is twice the length of the gap [6].

In the last years, computational intelligence techniques have gained popularity in pollen time series analysis [5], but their application in missing data imputation is less explored. Convolutional Neural Networks [5], Denoising Convolutional Auto-encoder [13], and k-Nearest Neighbours algorithm [14] are among the approaches used. Natural systems are physical (spatio-temporal) systems characterised by dominant non-linear structures that evolve over time (such as seasonality or climate variations) that are unknown. Identifying data tendencies connected to physics enables generalization for application across various fields [15–17]. Machine learning tools could be useful to repair corrupted or incomplete datasets, using the relevant spatio-temporal information directly from the data. Of these, the Singular Value Decomposition (SVD) is a data-driven multivariate method, useful for post-processing and handling data. The SVD, based on simple linear algebra, is the primary technique behind many dimensionality reduction methods, such as the Principal Component Analysis. The SVD method is able to recognise and extract the relevant spatio-temporal information directly from the data, removing noise and filtering out spatial redundancies, thus leading to dimensionality reduction. To address missing data, the Gappy SVD (GSVD) has been implemented, utilizing SVD properties to iteratively repair and reconstruct datasets. This algorithm has already been successfully applied to reconstruct fluid flow [16, 18] or oceanographic datasets [19], but it has never been tested on aerobiological datasets.

To assess imputation accuracy, simulation studies are conducted by generating missing data scenarios in complete datasets and comparing simulated and observed values. While simulation studies on environmental datasets have been widely explored (e.g. meteorological, hydrological, and air pollution data) [20–23], the challenges of imputation in aerobiological data remain less studied. Picornell et al. tested the ability of the different interpolation methods implemented in the "AeRobiology" R package, simulating random missing data in patterns of 3, 5, 7, and 10 consecutive days in different pollen seasonal periods (pre-season, pre-peak, peak, post-peak, and post-season) [1]. Navares et al. evaluated the performance of geographical imputation via Convolutional Neural Networks, generating 10, 20, and 30% of missing values in all periods, peak and off-peak season [5].

This paper introduces a novel implementation of the Gappy Singular Value Decomposition (GSVD) algorithm, a data-driven method, specifically tailored for the application to aerobiological datasets in this study. The imputation accuracy of this method was compared to a well-known statistical method, the moving mean algorithm.

## Materials and methods

### Aerobiological data

POLLnet is the aerobiological monitoring network of the National System for Environmental Protection (SNPA) of Italy, which aggregates aerobiological monitoring data measured by regions and provinces into a nationwide database (https://pollnet.isprambiente.it/). The network's monitoring follows the European Standard UNI EN 16868 2019, using Hirst-type volumetric samplers with a calibrated pump aspirating 10 l/min of air in 24 hours. Airborne particles are captured on a rotating metallic drum with an adhesive tape. The sampling drum is extracted every seven days, and the tape is cut into fragments corresponding to each monitoring day. These fragments are then examined under a microscope at 400× magnification by a specialized technician, and daily pollen grains are counted based on their morphological characteristics. The count is recorded as the number of pollen grains per cubic meter of air (p/m$^3$) [24, 25].

We constructed the aerobiological datasets using RStudio version 4.2.2 [26]. For the study purposes, we selected two monitoring stations representing different environments in the Northeast of Italy (Fig 1): VI1 in Vicenza, lowlands with continental climate, and BZ2 in Bolzano, mountains with alpine climate.

We downloaded daily pollen concentrations for the period 2018–2022 using the "pollnet" R package (https://rpubs.com/gbonafe/pollnet-data-extraction). The dataset is available at https://modelflows.github.io/modelflowsapp/airpollution/. *Alnus* and Poaceae pollens were considered for the analysis due to their different seasonality, temporal distribution, and load characteristics (as shown from the 2022 time series in Fig 1).

We computed the start and end dates of the season for each pollen time series using the 95-percentage method (start: 2.5%; end: 97.5%) from the "AeRobiology" R package [6, 27]. This method was solely used to define convenient periods within the pollen seasons to generate

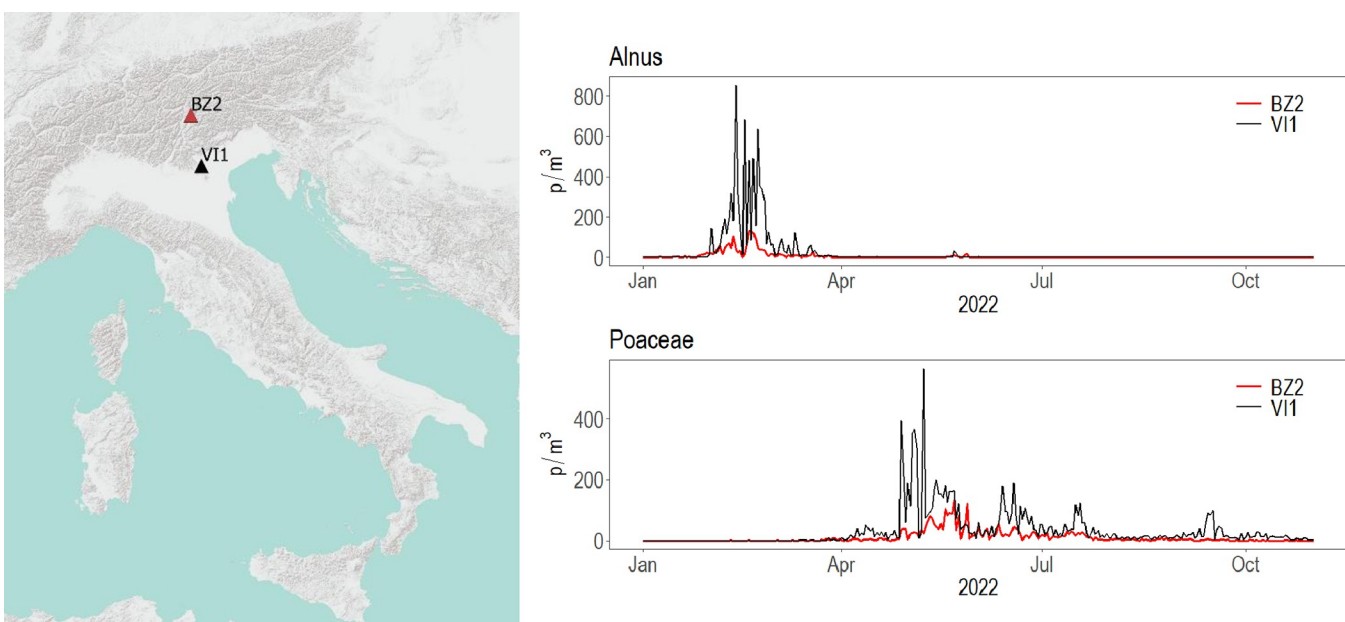

**Fig 1. Location of the selected monitoring stations in the Northeast of Italy and the respective pollen time series for the year 2022.** The map was produced using the QuickMapServices plugin (NextGIS, 2019) in QGIS software version 3.34.9 (QGIS Development Team. QGIS Geographic Information System. Open-Source Geospatial Foundation Project. http://qgis.org). The basemap used is ESRI Terrain (ESRI, Redlands, CA, USA). BZ2: Bolzano; VI1: Vicenza; p/m$^3$: pollen/cubic meter.

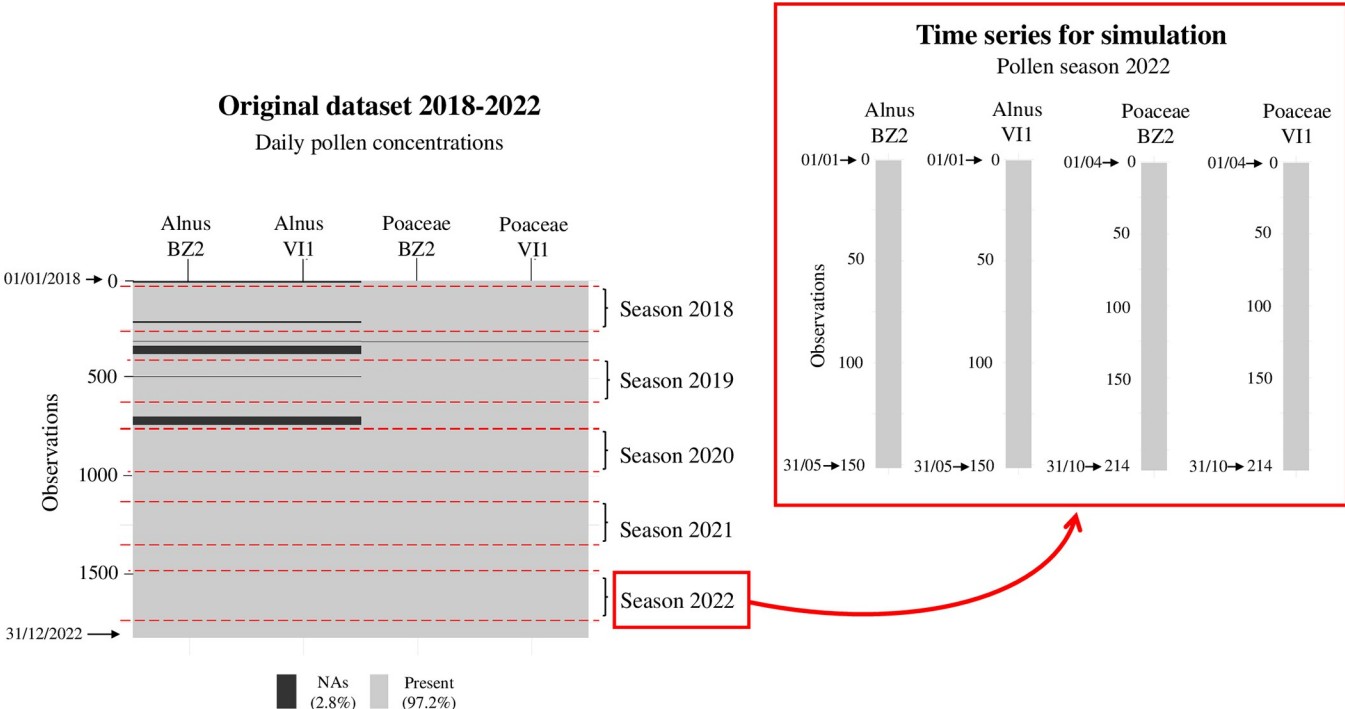

**Fig 2. Scheme depicting the original dataset of daily pollen concentrations for the period 2018–2022 and time series extracted for the simulation study.**
BZ2: Bolzano; VI1: Vicenza; NAs: missing data. Each season was obtained from the earlier start and later end day of the observed pollen seasons across the 2 monitoring stations: Season 2018 (start: 31/01/2018, *Alnus* BZ2; end: 17/09/2018, Poaceae VI1); Season 2019 (start: 11/02/2019, *Alnus* VI1; end: 17/09/2019, Poaceae VI1); Season 2020 (start: 30/01/2020, *Alnus* BZ2; end: 05/09/2020, Poaceae VI1); Season 2021 (start: 08/02/2021, *Alnus* VI1; end: 14/09/2021, Poaceae VI1); Season 2022 (start: 26/01/2022, *Alnus* BZ2; end: 08/10/2022, Poaceae VI1).

random missing data. Then, we examined missing data to identify the year with the most complete data coverage during the seasonal pollen period. The years from 2020 to 2022 showed no missing data at station VI1, whereas station BZ2 had complete data throughout the period (2018–2022). We chose to simulate the pollen season of the year 2022 to ensure a complete data series for the preceding years, thus guaranteeing the applicability of the data-driven method. Fig 2 shows the structure of the original dataset and the time series extracted for the simulation study.

Descriptive statistics of pollen concentrations at each station in the pollen season 2022 were calculated: mean ± Standard Deviation (SD), quartiles, coefficient of variation (CV = (SD/mean) × 100) (%), and duration of the pollen season.

As the start/end dates varied depending on the monitoring station, for each pollen, we considered a common seasonal period in the year 2022 by extending the season to the first day of the month on which the minimum start date occurred and to the last day of the month on which the maximum end date occurred. As a result, the period considered for imputation was 01/01/2022 to 31/05/2022 for *Alnus* and 01/04/2022 to 31/10/2022 for Poaceae.

## Methods of imputation investigated

We utilized the moving mean method of the "AeRobiology" R package as specifically developed for aerobiological datasets. The GSVD method used in this study was the algorithm originated from the ModelFLOWs-app (code available at https://modelflows.github.io/modelflowsapp/airpollution/), a novel software implementing modal decomposition

methods and hybrid machine learning tools to solve problems in complex nonlinear dynamical systems with application on patterns identification, data reconstruction, and data forecasting [16].

We initialised the GSVD algorithm assigning an initial value to the missing data. In this paper, the mean value of the time series (hereafter GSVD mean) and a linear interpolation between values of the time series (hereafter GSVD interp) were used for the initialisation. Then, SVD was applied to the initial dataset, as $X = U\Sigma V^T$, where the matrices U and V contain the modes (i.e. the spatio-temporal data decomposed by the Proper Orthogonal Decomposition mathematical approach) and the temporal coefficients, $()^T$ denotes the matrix transpose, and $\Sigma$ is the diagonal matrix containing the singular values of the matrix X. The first modes contain the physical modes related to the problem, while the rest are related to noise, spatial redundancies or to fit this initial guess. Retaining the first number N of modes, which can be tuned, one can approximate the database as $X^* = U^*\Sigma^*V^{T^*}$. The gaps of the original dataset were updated using the values of this approximation. Afterwards, SVD was applied again iteratively until the Mean Square Error (calculated as the ratio between the difference of the original and the reconstructed dataset and the total number of samples) of the gaps between two iterations is lower than a tolerance, set as $10^{-6}$. More information about the algorithm and the implementation can be found in Díaz-Morales et al. (2024) and Hetherington et al. (2023, 2024) [15–17].

## Simulation study

For each pollen type and station, we generated 12 simulation scenarios by combining 3 missing data proportions (5%, 10%, 25%) and 4 gap lengths (number of consecutive missing days: 3, 5, 7, 10 days). For each simulation scenario we obtained 100 simulated datasets. We randomly removed daily observed data from the complete pollen seasonal time series following the subsequent procedure (see Table 1):

i. calculation of the number of days within the pollen season corresponding to the total proportions of NAs of 5%, 10%, and 25%;

ii. calculation of the number of gaps for each gap length pattern (3, 5, 7, and 10 days) to approximate the total number of days with NAs from step i;

iii. implementation of the algorithm to randomly remove data iteratively 100 times in RStudio, setting the number of consecutive days and the number of gaps from steps i and ii without overlapping gaps.

As a result, we obtained a total of 48 simulations (12 scenarios x 2 stations x 2 pollens), each with 100 time series for imputation. An example of the NAs generation process and resulting dataset is reported in Fig 3.

The RStudio code for the generation of missing data and examples of gappy datasets are available at https://modelflows.github.io/modelflowsapp/airpollution/.

## Imputation and accuracy evaluation

To assess the accuracy of the imputation methods, we compared the reconstructed datasets to the observed time series, calculating the Root Mean Square Error (RMSE), i.e. the sum of the squared differences between the predicted and observed values divided by the total number of

**Table 1. Settings of simulation scenarios and the resulting percentages of NAs obtained from simulations.**

| Pollen | Season duration (days) | NAs simulation settings | | | | Resulting NAs | |
|---|---|---|---|---|---|---|---|
| | | % | Total days | Gap length (consequent days) | Number of gaps | Total days | % |
| *Alnus* | 151 (01 Jan—31 May) | 5 | 7.55 | 3 | 3 | 9 | 6 |
| | | | | 5 | 2 | 10 | 6.6 |
| | | | | 7 | 1 | 7 | 4.6 |
| | | | | 10 | 1 | 10 | 6.6 |
| | | 10 | 15.1 | 3 | 5 | 15 | 9.9 |
| | | | | 5 | 3 | 15 | 9.9 |
| | | | | 7 | 2 | 14 | 9.3 |
| | | | | 10 | 2 | 20 | 13.2 |
| | | 25 | 37.75 | 3 | 13 | 39 | 25.8 |
| | | | | 5 | 8 | 40 | 26.5 |
| | | | | 7 | 5 | 35 | 23.2 |
| | | | | 10 | 4 | 40 | 26.5 |
| Poaceae | 214 (01 Apr– 31 Oct) | 5 | 10.7 | 3 | 4 | 12 | 5.6 |
| | | | | 5 | 2 | 10 | 4.7 |
| | | | | 7 | 2 | 14 | 6.5 |
| | | | | 10 | 1 | 10 | 4.7 |
| | | 10 | 21.4 | 3 | 7 | 21 | 9.8 |
| | | | | 5 | 4 | 20 | 9.3 |
| | | | | 7 | 3 | 21 | 9.8 |
| | | | | 10 | 2 | 20 | 9.3 |
| | | 25 | 53.5 | 3 | 18 | 54 | 25.2 |
| | | | | 5 | 11 | 55 | 25.7 |
| | | | | 7 | 8 | 56 | 26.2 |
| | | | | 10 | 5 | 50 | 23.4 |

NAs: Missing data. The simulations are in total 12 for each pollen and station.

observations (N) (1).

$$RMSE = \sqrt{\frac{\sum_{i=1}^{N} (Predicted_i - Observed_i)^2}{N}} \tag{1}$$

As part of the "AeRobiology" R package, we executed the moving mean method using RStudio. We developed an algorithm to iteratively apply the "interpollen" function with "moving-mean" method to each column of individual datasets (pollen/station). After that, we merged the imputed dataset with the original corresponding pollen time series, and we implemented a function to iteratively calculate the RMSE between real data and the 100 replications of the simulated data. The final dataset contained the RMSE from all 100 simulations.

We implemented the GSVD algorithm in Python and ran it with Visual Studio Code version 1.86. As data-driven methods rely on extensive datasets to effectively capture data variability [28], we incorporated the 100 incomplete time series from each pollen, station, and simulation scenario into the original dataset including monitoring data spanning from 2018 to 2022. We studied different settings, changing the first initialisation of the values of the gaps and the number of modes, and evaluated the performance reconstruction of the gaps. Two imputation cases are shown in this paper for the sake of clarity, although other combinations

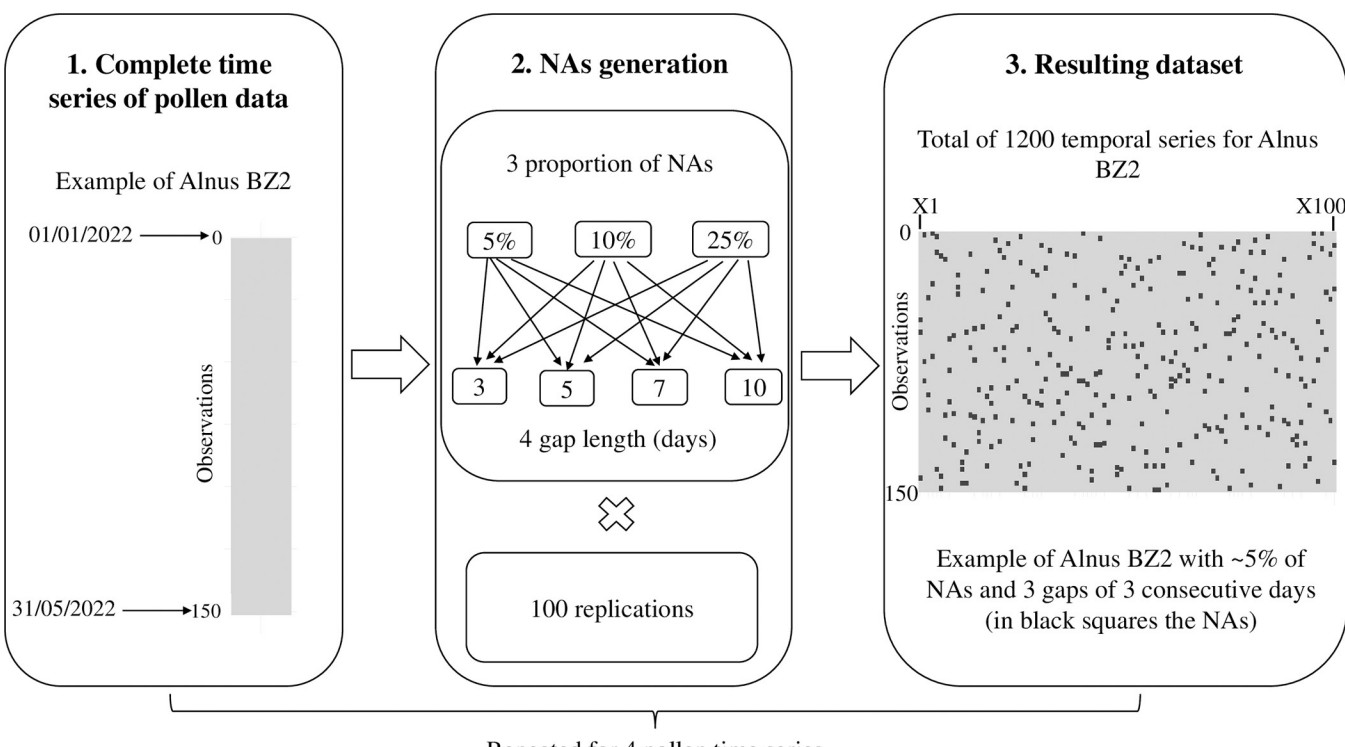

**Fig 3. Example of generation of missing values (NAs) and resulting dataset for *Alnus* BZ2.** BZ2: Bolzano.

showed similar results: the GSVD mean 5modes and the GSVD interp 10modes. At the end of each imputation, the algorithm calculated the RMSE for each repetition and for each pollen time series and extracted the results as a dataset.

Finally, we merged the RMSE from the different imputations, and then we calculated the median RMSE for each imputation method and each combination of NAs.

Besides this, we checked if the natural variability of the pollen may affect the imputation process, as reported by Picornell et al. [1]. Indeed, pollen distribution, load, and seasonality differ according to the environment, climate, and phenology of the plant. All these factors may impact the imputation accuracy. So, for each pollen time series, we calculated the Variation index (VIn), an indicator of variability in pollen concentrations between consecutive days, based on Picornell et al (2021) [1]:

i.  the moving mean and SD on consecutive 2 days within the pollen season;

ii.  the moving coefficients of variation (CV), as the ratio of the moving SD and moving mean;

iii.  the VIn, defined as the average of the moving CV over the pollen season.

Then, we related the median RMSE and the VIn using boxplots to explore the relation between imputation accuracy and pollen variability. Moreover, we employed multiple linear regression models stratified by pollen and monitoring station (M1: *Alnus* and BZ2 station; M2: *Alnus* and VI1 station; M3: Poaceae and BZ2 station; M4: Poaceae and VI1 station) to further explore this relation. The dependent variable was the RMSE from the 100 replications by all the simulations (total of 4800 time series), which we log-transformed to satisfy the normality assumption in linear regression. In addition, we applied a robust estimator of standard errors to relax the homoskedasticity assumption. Model covariates included the imputation method,

**Table 2. Descriptive statistics of pollen data in the pollen season 2022.**

| | Monitoring station | Mean ± SD (p/m³) | CV (%) | VIn (%) | 1st quartile (p/m³) | Median (p/m³) | 3rd quartile (p/m³) | Maximum (p/m³) | Duration of the season (days) |
|---|---|---|---|---|---|---|---|---|---|
| *Alnus* | BZ2 | 12.1±24.1 | 198.8 | 12.1 | 0.5 | 2.0 | 10.4 | 132.4 | 122 |
| | VI1 | 50.1±129.5 | 258.7 | 50.1 | 0.0 | 1.5 | 23.6 | 852.6 | 47 |
| Poaceae | BZ2 | 14.5±22.1 | 152.9 | 14.5 | 1.0 | 4.7 | 20.2 | 135.4 | 150 |
| | VI1 | 47.2±71.3 | 150.9 | 47.2 | 10.3 | 21.4 | 51.8 | 564.4 | 180 |

SD: Standard Deviation; CV: Coefficient of Variation; VIn: Variation Index; p/m³: pollen/cubic meter; BZ2: Bolzano; VI1: Vicenza.

proportion of NAs, and gap length. We exponentiated the regression coefficient β (Exp(β)) to provide an estimate of the relative change in RMSE.

## Results

### Pollen data description

Table 2 reports descriptive statistics of pollen observations in the pollen season 2022.

For both pollen types, the mean and SD presented higher values in the VI1 monitoring station than in BZ2. For *Alnus*, the duration of the pollen season was shorter in VI1 compared to BZ2, but pollen variability was higher. Instead, Poaceae showed a shorter pollen season in BZ2 than in VI1, but a higher variability in VI1 in terms of VIn.

### Performance analysis

No specific pattern resulted in the distribution of median RMSE values for pollen and station by imputation methods (Fig 4).

The variability in the distribution of median RMSE was lower at the BZ2 monitoring station (*Alnus*: from 0.6 to 9.5 p/m³; Poaceae: from 0.9 to 8.2 p/m³) and higher at the VI1 monitoring station (*Alnus*: from 1.5 to 56.5 p/m³; Poaceae: from 4.1 to 27.6 p/m³).

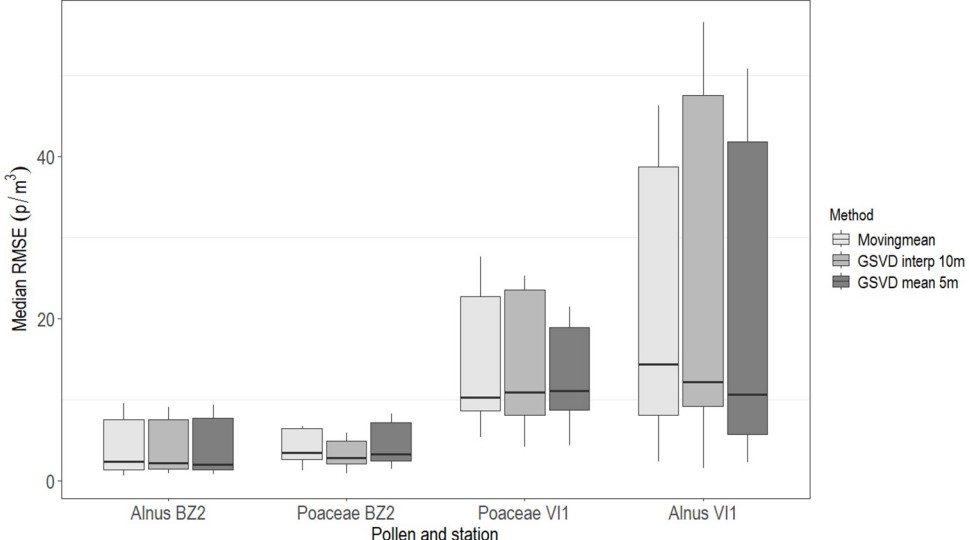

**Fig 4. Distribution of the median Root Mean Square Error (RMSE) values for pollen/station by imputation method.** BZ2: Bolzano; VI1: Vicenza; GSVD: Gappy Singular Value Decomposition; p/m³: pollen/cubic meter. Each box represents the distribution of the median RMSE from the 12 simulations.

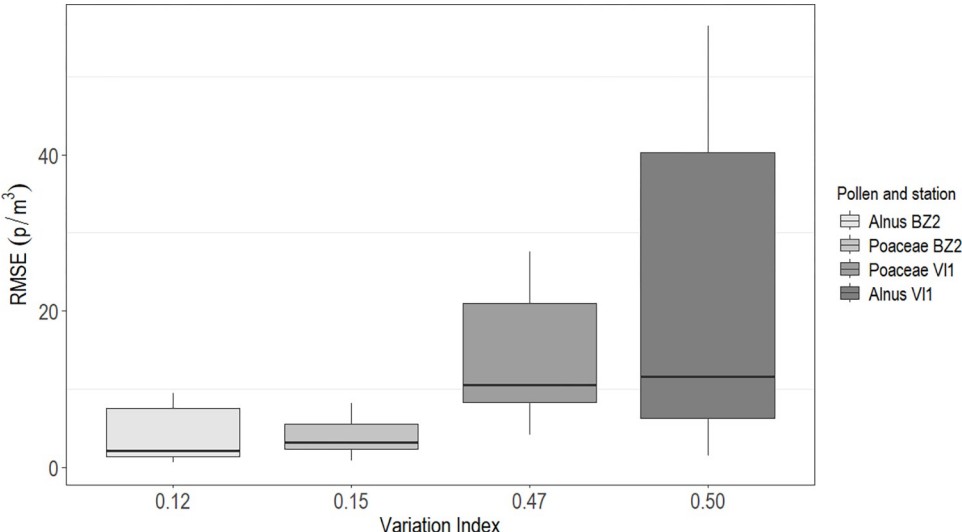

**Fig 5. Distribution of median Root Mean Square Error (RMSE) for the Variation Index by pollen/station.** BZ2: Bolzano; VI1: Vicenza; p/m³: pollen/cubic meter. Each box represents the distribution of the median RMSE from the 12 simulations imputed with the 3 methods.

When examining the relationship between the median RMSE values and VIn (Fig 5), a trend of increasing RMSE with higher VIn values emerged. Moreover, higher variability in the distribution of median RMSE values increased with higher VIn values.

Based on the results of the multiple linear regression models, there was large variability in imputation accuracy across the methods investigated, and none of them outperformed the others when adjusting for the simulation scenario (Table 3).

Moreover, no consistency was found within the GSVD imputation method, even showing contrasting results as in model M3. There was instead a consistent association between the

**Table 3. Association estimates (Exp(β) representing ratios of geometric means) with 95%CI between the Root Mean Square Error (RMSE) and covariates (imputation method, % of NAs, and gap length).**

|  | M1 (*Alnus*, BZ2) | M2 (*Alnus*, VI1) | M3 (*Poaceae*, BZ2) | M4 (*Poaceae*, VI1) |
|---|---|---|---|---|
|  | Exp(β) (95%CI) | Exp(β) (95%CI) | Exp(β) (95%CI) | Exp(β) (95%CI) |
| Imputation method: |  |  |  |  |
| Moving mean | Ref. | Ref. | Ref. | Ref. |
| GSVD interp 10m | 1.20 (1.05–1.38) | 1.00 (0.81–1.24) | 0.88 (0.80–0.97) | 0.94 (0.85–1.04) |
| GSVD mean 5m | 1.03 (0.89–1.18) | 1.13 (0.93–1.38) | 1.10 (1.00–1.21) | 0.92 (0.83–1.00) |
| % of NAs: |  |  |  |  |
| 5 | Ref. | Ref. | Ref. | Ref. |
| 10 | 2.12 (1.80–2.49) | 3.76 (2.93–4.84) | 1.94 (1.72–2.19) | 1.80 (1.60–2.01) |
| 25 | 5.85 (5.11–6.69) | 9.86 (8.03–12.10) | 3.84 (3.49–4.22) | 3.74 (3.39–4.13) |
| Gap length (days): |  |  |  |  |
| 3 | Ref. | Ref. | Ref. | Ref. |
| 5 | 0.92 (0.79–1.08) | 1.14 (0.88–1.46) | 0.84 (0.75–0.94) | 0.90 (0.80–1.02) |
| 7 | 0.65 (0.56–0.76) | 0.49 (0.39–0.63) | 0.94 (0.85–1.03) | 0.91 (0.82–1.01) |
| 10 | 0.76 (0.66–0.88) | 0.74 (0.60–0.92) | 0.77 (0.69–0.85) | 0.78 (0.71–0.87) |

BZ2: Bolzano; VI1: Vicenza; CI: Confidence Interval; GSVD: Gappy Singular Value Decomposition; NAs: missing values; Ref.: reference category.

simulation settings and imputation accuracy. In fact, the RMSE increased with an increasing proportion of NAs across all models. Notably, the RMSE was 4 to 10 times higher when NAs were set to 25%, compared to the reference of 5%. On the contrary, the RMSE decreased with gap length, showing minimum values at 7 days (M1 and M2) and 10 days (M3 and M4).

## Discussion and conclusions

A simulation study was conducted to compare the imputation accuracy of two methodologies, applying and evaluating for the first time the GSVD method to aerobiological datasets. Promising results emerged, demonstrating a similar performance of GSVD in comparison to the well-established moving mean method of the "AeRobiology" R package. However, it was found that both the inherent variability in observed pollen concentrations and the pattern of missing data had a more substantial impact on imputation accuracy within aerobiological datasets than the interpolation method applied. These findings contribute to filling the gap of knowledge in this field, considering the limited number of simulation studies conducted on pollen time series [1, 5, 13].

We compared univariate and multivariate methods of interpolation specifically focusing on aerobiological datasets. Previous simulation studies on other types of environmental data (e.g. hydrological, meteorological, air quality) have favoured multivariate methods, leveraging information from other temporal series, over univariate methods, which rely solely on the data series itself [20–22]. On one hand, we used the moving mean algorithm from the "AeRobiology" R package as univariate method, which was specifically designed for aerobiological datasets. This algorithm was identified within the package as the interpolation method with better performance, attributed to its reduced sensitivity to data availability, time series length, and fluctuations in pollen concentrations across consecutive days [1]. Its simplicity and increasing usage in aerobiological studies underscores its relevance and effectiveness in reconstructing time series. On the other hand, we used the GSVD algorithm as multivariate method, first evaluating its performance on aerobiological datasets. The potential of this method lies in its ability to reduce data dimensionality through data-driven decomposition, identifying the main data patterns related to physics without requiring any assumptions [15–17]. This makes it a promising tool for dataset reconstruction, as evidenced by its strong generalization capabilities across different types of data. However, more applications of the GSVD method on aerobiological data are needed to evaluate its effectiveness across diverse pollen types and environmental conditions. Indeed, the GSVD performance resulted similar to that of the statistical approach, with both methods exhibiting similarly unsatisfactory imputation accuracy in some settings. Moreover, the comparison of these two methods in our study revealed insights into the various factors influencing imputation performance. It suggested that the specific characteristics and requirements of the dataset may play a significant role in determining the most suitable interpolation approach.

The challenge of imputing missing data in aerobiological datasets is compounded by the complexity of plant phenology and pollen diffusion and advection mechanisms. Beyond their non-normal statistical distribution, each pollen type is influenced by local environmental and climatic conditions, resulting in differences in quantity, seasonality, and daily concentrations patterns [1]. Meteorological factors, particularly temperature and precipitation, are widely acknowledged to have the greatest influence on pollen variability, affecting both phenological phases and pollen behaviour in the atmosphere [29, 30]. Hence, the same pollen type may exhibit different distribution curves depending on the location characteristics [1]. Such variability has been related to decreased accuracy in imputation, as wider concentration ranges between consecutive days heighten the likelihood of errors during the imputation process [1].

This association has been observed in other environmental data as well [31]. Our findings align with Picornell et al. (2021) [1], indicating that higher variability in concentration (VIn) resulted in less accurate imputation results, both in terms of values and range of variability of the imputation error. Notably, measurements from the Vicenza station showed greater variability, likely attributable to the effect of continental climate characteristics of the lowlands on pollen, subjected to significant thermal fluctuations compared to alpine regions. Additionally, *Alnus* pollen generally displayed higher VIn values compared to Poaceae pollen, likely due to significant variability over a shorter season duration.

Besides the pollen type and location of the monitoring station, the pattern of missing data had the most substantial impact on imputation accuracy in our study. We generated missing data by introducing fixed consecutive-day gaps at various percentages in aerobiological datasets. The results showed a trend of increasing imputation error with higher percentages of NAs, regardless of the pollen/location. Our results align with the findings of Junger et al. (2015) concerning air pollution data, indicating that 5% of missing data yields satisfactory results, but accuracy decreases with more than 10% missing data [21]. In contrast, one study found opposing trends with increasing percentages of missing data for different meteorological variables [31], while another study observed no specific trend between missing data percentage and imputation error in aerobiological databases [5].

Regarding the gap length, our findings differ from those of Picornell et al. (2021) [1], as we observed that interpolation error decreases with longer gap lengths, depending on the pollen type. Specifically, the imputation error was minimum in datasets with gaps of 7 consecutive days for *Alnus*, and with gaps of 10 days for Poaceae, compared to gaps of 3 days. Despite the higher possibility of abrupt variations in longer gaps [1], the observed decrease in error with longer gaps can be attributed to the smoothing effect of interpolation. This effect leads to a reduction in the likelihood of generating peaks through interpolation, thereby minimizing errors. Notably, this effect appears to be more pronounced for pollens with wider season duration and less variability, as seen for Poaceae. The abundance of pollen-producing plants within this family, comprising over 120 genera in Italy, leads to high atmospheric pollen levels persisting over extended periods, thereby reducing day-to-day variability and smoothing peaks in pollen concentrations.

In conclusion, missing data resulting from manual measurement are common in aerobiological datasets [1, 5, 21]. Therefore, imputation remains the best solution for dealing with incomplete datasets and is useful for improving aerobiological analysis [1, 32]. In fact, even small gaps can distort estimates in environmental epidemiology or climatological studies [13]. Omitting to address missing data can result in significant errors in analysing pollen time series, which in turn can affect the definition of pollen seasonality [1, 7, 8]. We introduced and tested a novel method for missing data imputation in aerobiological research, demonstrating comparable performance to the moving mean method in data reconstruction. Both methods yielded favourable results, with the moving mean method being the simpler option. However, the imputation error remained unacceptable for certain pollen types and missing data scenarios. Additional research is required to investigate the application of the GSVD method across diverse pollen types and environmental conditions to draw a definitive conclusion. Furthermore, incorporating meteorological data into pollen datasets should be considered to improve imputation accuracy. Finally, there is a need to improve current imputation methods and develop more reliable techniques specifically tailored to pollen data, aiming to minimize the impact of temporal variability in pollen concentrations on imputation error.

## Author Contributions

**Conceptualization:** Soledad Le Clainche.

**Data curation:** Sofia Tagliaferro, Adrián Corrochano.

**Formal analysis:** Sofia Tagliaferro, Adrián Corrochano.

**Funding acquisition:** Alessandro Marcon.

**Investigation:** Sofia Tagliaferro, Adrián Corrochano, Alessandro Marcon, Soledad Le Clainche.

**Methodology:** Sofia Tagliaferro, Adrián Corrochano, Pierpaolo Marchetti, Alessandro Marcon, Soledad Le Clainche.

**Project administration:** Alessandro Marcon.

**Software:** Adrián Corrochano.

**Supervision:** Pierpaolo Marchetti, Alessandro Marcon, Soledad Le Clainche.

**Visualization:** Sofia Tagliaferro, Pierpaolo Marchetti, Alessandro Marcon.

**Writing – original draft:** Sofia Tagliaferro.

**Writing – review & editing:** Sofia Tagliaferro, Adrián Corrochano, Pierpaolo Marchetti, Alessandro Marcon, Soledad Le Clainche.

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
