## [Decision Letter · Decision Letter 0]

9 Sep 2024

PONE-D-24-22455A new method based on physical patterns to impute aerobiological datasetsPLOS ONE

Dear Dr. Marcon,

Thank you for submitting your manuscript to PLOS ONE. After careful consideration, we feel that it has merit but does not fully meet PLOS ONE’s publication criteria as it currently stands. Therefore, we invite you to submit a revised version of the manuscript that addresses the points raised during the review process.

**Your manuscript has been evaluated by three reviewers, and their comments are appended below and in the attached file.** **The reviewers have primarily requested clarification of some aspects of your study, however, please address in detail the comment from Reviewer 2 regarding calculation of the start and end dates of the pollen season for each pollen type. Please ensure you address each of the reviewers' comments when revising your manuscript.** **We note that one or more reviewers has recommended that you cite specific previously published works. As always, we recommend that you please review and evaluate the requested works to determine whether they are relevant and should be cited. It is not a requirement to cite these works. We appreciate your attention to this request.**

We look forward to receiving your revised manuscript.

Kind regards,

Hugh Cowley

Staff Editor

PLOS ONE

**Journal Requirements:**

This research has received grants from the European Union through the Italian Ministry of University and Research under the ESF REACT-EU Green and Innovation funding programme (Ministerial Decree 1061/2021) and the NextGenerationEu funding programme (Ministerial Decree 737/2021). Article processing charges were supported by the special fund at the University of Verona dedicated to Open Access publications. The funder had no role in study design, data collection and analysis, decision to publish, or preparation of the manuscript.

Reviewers' comments:

Reviewer's Responses to Questions

**Comments to the Author**

1. Is the manuscript technically sound, and do the data support the conclusions?

Reviewer #1: Yes

Reviewer #2: Yes

Reviewer #3: Yes

2. Has the statistical analysis been performed appropriately and rigorously? 

Reviewer #1: Yes

Reviewer #2: Yes

Reviewer #3: Yes

3. Have the authors made all data underlying the findings in their manuscript fully available?

Reviewer #1: Yes

Reviewer #2: Yes

Reviewer #3: Yes

4. Is the manuscript presented in an intelligible fashion and written in standard English?

Reviewer #1: Yes

Reviewer #2: Yes

Reviewer #3: Yes

5. Review Comments to the Author

**Reviewer #1:** This study (A new method based on physical patterns to impute aerobiological datasets) includes an analysis of the effectiveness of Gappy Singular Value Decomposition (GSVD) in aerobiological datasets. It addresses a clear and evident need in the field of aerobiology: the lack of data on pollen concentration on some days. I would like to congratulate the authors of the paper because it is very well written. Thus, I consider that this paper, after minor corrections, is suitable for publication in PLOS ONE. Some minor suggestions are listed below.

L177-179: Use roman numerals i and ii without italics.

L322: While reading this paragraph, as a researcher, I would opt for the "simpler" method. Perhaps add that more studies applying GSVD on different pollen types and environments are needed to reach a more generalized conclusion.

L327: Emphasize the importance of linking pollen data with meteorological data, as has been observed, for example, in this study https://doi.org/10.1016/j.scitotenv.2021.145426

**Reviewer #2: **Dear Authors,

My review is as follows.

In addition, it is attached as a pdf file for keeping its format.

Reviewer

Tagliaferro, S., Corrochano, A., Marchetti, P., Marcon, A., Le Clainche, S., 2024: A new method based on physical patterns to impute aerobiological datasets. PLOS ONE;

The authors conducted a novel simulation study to evaluate the effectiveness of Gappy Singular Value Decomposition (GSVD), as a data-driven approach, comparing it with the moving mean interpolation, as a statistical approach. However, high variability in pollen concentrations and the increasing number of missing data and the increasing lack of those around the sample mean negatively affected imputation accuracy.

My comments are as follows.

Comments:

• Line 123: You used the 95% method (start: 2.5th percentile; end: 97.5th percentile) when calculating the start and end dates of the pollen season for each pollen type.

However, if pollen concentrations and within-season distribution vary by year, such an approach is likely to contract or expand season duration independently from climate change. For example, if the API was 2000 for a given year, the start of the pollen season would be when the cumulative concentration reached 50; but if the API for the same location was 5000 for the following year, then the pollen season would not start until the cumulative concentration reached 125. Use of this system to mark the start and end of the pollen season would mask any climate-related or temperature-related changes associated with pollen season duration.

Therefore, rather than using start and end percentages, to determine the pollen season, it is widely used the first (last) date on which at least 1 pollen grain m−3 of air is recorded and at least 5 consecutive (preceding) days also show 1 or more pollen grains m−3 (Makra et al., 2012).

Reference

Makra, L., Matyasovszky, I., Bálint, B., 2012: Association of allergic asthma emergency room visits with the main biological and chemical air pollutants. Science of the Total Environment, 432, 288-296. doi:10.1016/j.scitotenv.2012.05.088

• Line 136: Graminaceae is the former name of the grass family. I recommend using Poaceae everywhere in the manuscript instead.

• Line 247: correctly: “For both pollen types”, instead of “For both pollens”;

• Line 341: correctly: “with”, instead of “whit”;

Reviewer

**Reviewer #3: **The manuscript entitled ”A new method based on physical patterns to impute aerobiological datasets” by Tagliaferro et al. presents a novel method to handle missing values in the monitorization of bioaerosols, in comparison to the current method of managing missing data, while also analyzing which factors influence the accuracy of these methods. For this, the authors use pollen data from two different pollen species from two different monitoring stations. The topic of the paper is highly relevant in the context of worldwide increasing allergy prevalence and the influence of climate change on the distribution of respiratory allergens. The paper is overall well written explaining the methods employed for the data comparison.

There are a few minor comments regarding the manuscript

1. Could the authors add some information regarding what pollen genera are comprised within the examined Graminaceae pollen and discuss how this could influence the accuracy of the presented methods since they mention a lower variability in this pollen type and a longer pollen season?

2. Line 120: The authors mention analyzing Alnus and Poaceae pollen, whereas throughout the manuscript they use Graminaceae pollen

3. Line 341 – typo probably ”with” rather than ”whit”

6. PLOS authors have the option to publish the peer review history of their article (what does this mean?). If published, this will include your full peer review and any attached files.

Reviewer #1: No

Reviewer #2: No

Reviewer #3: No

---

## [Author Response · Author response to Decision Letter 0]

10 Oct 2024

The answers to Reviewers' questions are available in the attached file "Response to Reviewers".

---

## [Decision Letter · Decision Letter 1]

5 Nov 2024

A new method based on physical patterns to impute aerobiological datasets

PONE-D-24-22455R1

Dear Dr. Marcon,

We’re pleased to inform you that your manuscript has been judged scientifically suitable for publication and will be formally accepted for publication once it meets all outstanding technical requirements.

Kind regards,

Rajeev Singh

Academic Editor

PLOS ONE

Additional Editor Comments (optional):

Reviewers' comments:

Reviewer's Responses to Questions

**Comments to the Author**

1. If the authors have adequately addressed your comments raised in a previous round of review and you feel that this manuscript is now acceptable for publication, you may indicate that here to bypass the “Comments to the Author” section, enter your conflict of interest statement in the “Confidential to Editor” section, and submit your "Accept" recommendation.

Reviewer #1: All comments have been addressed

Reviewer #2: All comments have been addressed

Reviewer #3: All comments have been addressed

2. Is the manuscript technically sound, and do the data support the conclusions?

Reviewer #1: Yes

Reviewer #2: Yes

Reviewer #3: Yes

3. Has the statistical analysis been performed appropriately and rigorously? 

Reviewer #1: Yes

Reviewer #2: Yes

Reviewer #3: Yes

4. Have the authors made all data underlying the findings in their manuscript fully available?

Reviewer #1: Yes

Reviewer #2: Yes

Reviewer #3: Yes

5. Is the manuscript presented in an intelligible fashion and written in standard English?

Reviewer #1: Yes

Reviewer #2: Yes

Reviewer #3: Yes

6. Review Comments to the Author

Reviewer #1: The content relating to the previous review has been updated and therefore this paper is suitable for publication in its current form.

Reviewer #2: Dear Authors,

I accept your answer to my comments and suggest to publish the revised version of your manuscript as it is.

Reviewer

Reviewer #3: Considering the authors' adequate and complete responses to all my comments, I recommend the article to be accepted for publication.

7. PLOS authors have the option to publish the peer review history of their article (what does this mean?). If published, this will include your full peer review and any attached files.

Reviewer #1: No

Reviewer #2: No

Reviewer #3: No

---

## [Editor Report · Acceptance letter]

8 Nov 2024

PONE-D-24-22455R1 

PLOS ONE

Dear Dr. Marcon, 

I'm pleased to inform you that your manuscript has been deemed suitable for publication in PLOS ONE. Congratulations! Your manuscript is now being handed over to our production team.

Kind regards, 

on behalf of

Dr. Rajeev Singh 

Academic Editor

PLOS ONE